# SpectR: Dynamically Composing LM Experts with Spectral Routing

**William Fleshman & Benjamin Van Durme**
Johns Hopkins University
`will.fleshman@jhu.edu`

## Abstract

Training large, general-purpose language models poses significant challenges. The growing availability of specialized *expert* models, fine-tuned from pretrained models for specific tasks or domains, offers a promising alternative. Leveraging the potential of these existing expert models in real-world applications requires effective methods to select or merge the models best suited for a given task. This paper introduces SPECTR, an approach for dynamically composing expert models at each time step during inference. Notably, our method requires no additional training and enables flexible, token- and layer-wise model combinations. Our experimental results demonstrate that SPECTR improves routing accuracy over alternative training-free methods, increasing task performance across expert domains.[1]

## 1 Introduction

Language models (LMs) have increasingly become more capable in recent years with model size being an important factor in scaling performance (Kolachina et al., 2012; Hestness et al., 2017; Kaplan et al., 2020; Hoffmann et al., 2022). Unfortunately, large models are inherently more resource-intensive, motivating the development of efficiency-boosting strategies such as knowledge distillation (Hinton et al., 2015; Gou et al., 2021), quantization (Shen et al., 2020; Kim et al., 2021; Wu et al., 2022; Dettmers & Zettlemoyer, 2023), and pruning (Fang et al., 2023; Ma et al., 2023). Architecting models as a Mixture-of-Experts (MoEs) has also become popular for state-of-the-art open-source LMs such as the DeepSeek (DeepSeek-AI et al., 2024) Mixtral (Jiang et al., 2024), and Qwen (Qwen et al., 2025) model families. MoEs gain efficiency by learning to activate a smaller subset of parameters for each input, reducing computation while allowing for larger model sizes (Jacobs et al., 1991; Fedus et al., 2022).

Simultaneously, parameter efficient fine-tuning (Mangrulkar et al., 2022) methods such as adapter-tuning (Bapna & Firat, 2019; Houlsby et al., 2019), prefix-tuning (Li & Liang, 2021), and prompt-tuning (Lester et al., 2021; Liu et al., 2022) have emerged to allow efficient customization of pretrained models for expert-level performance in new domains or tasks. Low-rank adaptation (LoRA) (Hu et al., 2022) is one of the most widely adopted adapter-tuning approaches, resulting in the proliferation of thousands of expert models openly available in repositories such as huggingface (Wolf et al., 2020). Given the abundance of these experts, many strategies have been proposed for merging models to improve multi-task capability (Ilharco et al., 2023; Yadav et al., 2023; Yu et al., 2024; Stoica et al., 2025).

Model *MoErging* pairs these merging strategies with routing mechanisms akin to MoEs (Yadav et al., 2024). Unfortunately, the majority of existing MoErging methods require training data for learning to route (Pfeiffer et al., 2021; Shnitzer et al., 2023; Huang et al., 2024; Tang et al., 2024) or custom training procedures for enabling adapter compatibility or gathering activation statistics (Chronopoulou et al., 2023; Diao et al., 2023; Belofsky, 2024; Fleshman et al., 2024). These constraints motivate the development of *training-free* MoErging approaches which require no data and allow for the use of externally sourced LoRA experts.

---

[1]Code available at `https://github.com/wfleshman/spectr`

One training-free option is the use of $\mu$-routing, which forgoes selecting specific experts and instead routes every input to all models (Caccia et al., 2023; Ostapenko et al., 2024). While dense linear combinations of experts can be successful (Wortsman et al., 2022; Chronopoulou et al., 2023; Ilharco et al., 2023), merging several LoRAs is especially problematic due to interference among adapters (Ortiz-Jimenez et al., 2023; Tang et al., 2024; Stoica et al., 2025).

Ostapenko et al. (2024) proposed Arrow routing, which uses the singular value decomposition of LoRA adapters for crafting a prototype vector per expert. These prototypes are then used to score input vectors by measuring the magnitude of their dot product, routing then to the top-k experts with the highest score. We recognize Arrow as a significant contribution, while developing an improvement we refer to as Spectral Routing (SPECTR) that addresses limitations in existing training-free methods such as Arrow (Figure 1). Specifically, we:

- Identify and confirm weaknesses in existing training-free routing strategies;

- Propose SPECTR, our approach designed to explicitly address these challenges;

- Measure the impact of adapter rank and task similarity on routing effectiveness, demonstrating improved routing accuracies with SPECTR across 4 LMs;

- Quantify the impact of different routing strategies on multi-task performance, with SPECTR increasing accuracy by up to 15% over alternatives; and

- Discuss trade-offs between SPECTR and other training-free methods.

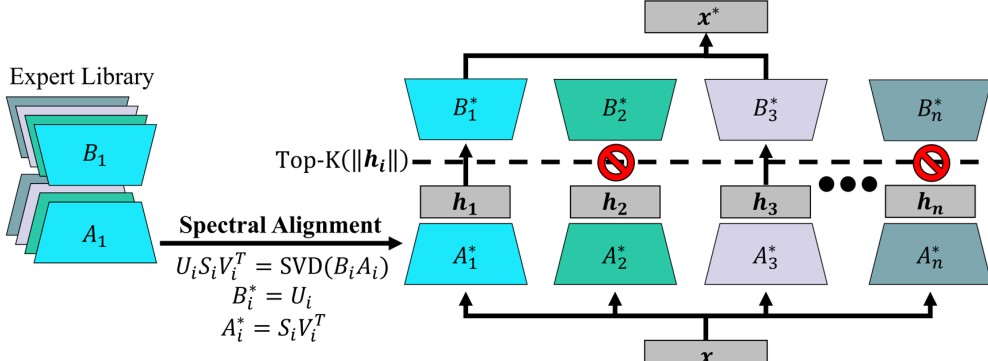

Figure 1: SPECTR uses the SVD to transform adapters into equivalent representations capable of measuring the compatibility of new inputs without relying on expert prototypes or routing networks. Vectors ($\mathbf{x}$) are projected into low-rank representations ($\mathbf{h}_i$) using the eigenvectors ($A_i^*$) of each adapter covariance matrix. The length of these representations measures the alignment of the input with directions of maximum variation induced by the expert in the space of possible inputs. Routing continues for the top-k compatible experts.

## 2 Background and Motivation

### 2.1 Adapters as Expert Models

**Adapters** enable parameter efficient fine-tuning (PEFT) of existing models to new tasks or domains by adding a small number of trainable parameters to existing models (Bapna & Firat, 2019; Houlsby et al., 2019). Low-rank adaptation (LoRA) is one of the most popular and effective adapter-tuning approaches, accomplished by adding a low-rank matrix product to each adapted layer (Hu et al., 2022). LoRA's popularity is partly due to technologies such as Mangrulkar et al. (2022)'s peft library and (Wolf et al., 2020)'s huggingface model repository, resulting in wide availability of expert LoRA models trained for a diverse set of tasks and domains (Brüel-Gabrielsson et al., 2024). In this work, we use LoRA adapters to derive several *expert* models, each capable of superior performance on their specific task. We

explore methods for dynamically selecting and combining these experts at inference time with no prior knowledge of the applicability or compatibility of individual expert models.

**Mixture of Experts** (MoEs) efficiently scales the capacity of LLMs by only activating a subset of the model parameters for each input (Jacobs et al., 1991; Fedus et al., 2022). These models are jointly trained without explicitly assigning individual tasks to each expert. The MoE approach has gained significant use in practice, resulting in multiple state-of-the-art language models in recent years (DeepSeek-AI et al., 2024; Jiang et al., 2024; Qwen et al., 2025). The success of MoE has motivated the development of techniques for mixing task-specific experts such as Ye et al. (2022)'s Task-level MoE and several adapter-based expert approaches (Pfeiffer et al., 2021; Wang et al., 2022; Caccia et al., 2023; Ponti et al., 2023; Fleshman et al., 2024; Huang et al., 2024; Zadouri et al., 2024). Unlike these existing works, we require zero data or training to select and combine our mixture of task-specific experts.

**Task-Arithmetic** isolates parameters responsible for a specific task by taking the difference between model weights before and after fine-tuning on the task (Ilharco et al., 2023; Fleshman & Van Durme, 2024). Fleshman & Van Durme (2024) demonstrate that these differences can be decomposed into LoRA adapters using the singular value decomposition (SVD). We note that these arithmetic-based adapters allow for arbitrary expert models to be used for adapter mixing, but our experiments in this work focus only on traditional LoRA models.

## 2.2 Adapter Selection and Merging

**Model merging** is an established technique for deriving a single multi-task model from individual task-specific models (Matena & Raffel, 2022; Wortsman et al., 2022; Stoica et al., 2024). One popular approach for model merging is to create a linear combination of the expert model weights (Wortsman et al., 2022; Chronopoulou et al., 2023; Ilharco et al., 2023; Fleshman et al., 2024). While simple, these weighted averages can be less effective when using LoRA experts, especially as the number of experts grows (Tang et al., 2024; Stoica et al., 2025). Tang et al. (2024) hypothesizes that LoRA adapters are less disentangled than full-rank fine-tuned models, causing deleterious interference when combined. Chronopoulou et al. (2023) and Fleshman et al. (2024) observed that LoRA averaging worked best when adapters for different tasks were equally initialized, an approach inapplicable when using experts from multiple sources. These challenges have led to many techniques for mitigating adapter interference (Ortiz-Jimenez et al., 2023; Yadav et al., 2023; Tang et al., 2024; Stoica et al., 2025; Yu et al., 2024). Ortiz-Jimenez et al. (2023) and Tang et al. (2024) suggest fine-tuning strategies to discourage entanglement during expert training. Yadav et al. (2023) propose TIES, which resolves adapter interference through an election process during merging. Stoica et al. (2025)'s KnOTS method performs an SVD over the set of adapters and merges them in the new shared subspace. Yu et al. (2024)'s DARE randomly drops and rescales parameters to create sparse approximations of experts. Our approach is compatible with most post-training strategies such as DARE, KnOTS, or TIES, but we find simple averaging with a sparse selection of experts reduces complexity and works well in our experiments.

**Routing** is the process of selecting relevant experts for a given task or query (Yadav et al., 2024). Yadav et al. (2024) refer to the combination of routing and merging as *MoErging* to highlight the similarity to MoEs while distinguishing methods using existing individual experts. We adopt their MoErging taxonomy which categorizes routing by the type of dataset used to learn routing, input and depth granularity, and the expert selection and aggregation mechanisms (Yadav et al., 2024). For example, $\mu$-routing does not require training data because it aggregates all experts as a uniform linear combination for all queries and model layers (Caccia et al., 2023; Ostapenko et al., 2024). In contrast, Chronopoulou et al. (2023) and Fleshman et al. (2024) perform clustering over dense representations of training data and route new queries to a mixture of experts represented by similar clusters. Our work focuses on situations where there is no data available for learning to route. Therefore, we include $\mu$-routing as one of the appropriate baselines for our MoErging experiments.

## 2.3  Problem Setting

We study training-free model MoErging methods using experts derived from standard LoRA training procedures (Hu et al., 2022; Ostapenko et al., 2024; Yadav et al., 2024). We assume access to $T$ task-specific LoRAs for the same LM architecture without access to any data associated with their training. For a query $q_t$ related to task $t$, we would like to select and merge a small number of experts (including the expert trained for $t$) capable of addressing $q_t$. We choose this setting to enable the usage of large and distinctly trained expert repositories that may not include associated data. We use LoRA for parameter efficiency (Hu et al., 2022) and choose sparse expert selection to minimize interference while increasing the chances of selecting the correct expert under imperfect routing (Tang et al., 2024; Yadav et al., 2024).

## 2.4  Arrow Routing

Arrow routing (Ostapenko et al., 2024) is the only existing work covered by the MoErging taxonomy (Yadav et al., 2024) that meets the requirements of our problem setting. Ostapenko et al. (2024) interpret the routing matrix from a standard MoE model as a collection of expert *prototypes* and construct their own prototypes using singular value decomposition (SVD). Recall that a rank-$r$ LoRA for task $t$ maps an input $\mathbf{x} \in \mathbb{R}^{d_{\text{in}}}$ to an output $\mathbf{x}^* \in \mathbb{R}^{d_{\text{out}}}$ with:

$$\mathbf{x}^* = W\mathbf{x} + B_t A_t \mathbf{x}, \tag{1}$$

where $W \in \mathbb{R}^{d_{\text{out}} \times d_{\text{in}}}$ are the original layer weights, and $B_t \in \mathbb{R}^{d_{\text{out}} \times r}$ and $A_t \in \mathbb{R}^{r \times d_{\text{in}}}$ are the low-rank LoRA matrices (Hu et al., 2022)[2]. The SVD of the LoRA product:

$$U_t, S_t, V_t = \text{SVD}(B_t A_t), \tag{2}$$

produces the matrix containing the left singular vectors $U_t \in \mathbb{R}^{d_{\text{out}} \times r}$, the diagonal matrix of singular values $S_t \in \mathbb{R}^{r \times r}$, and the matrix of right singular vectors $V_t \in \mathbb{R}^{d_{\text{in}} \times r}$ such that:

$$U_t S_t V_t^T = B_t A_t. \tag{3}$$

The right singular vectors are eigenvectors of the LoRA parameter covariance matrix $(B_t A_t)^T B_t A_t$ and represent orthogonal directions of maximum variation induced by the LoRA expert in the space of input vectors $\mathbf{x}$. The top eigenvector $\mathbf{v_t} = V_t[:, 0]$ satisfies the following equation over possible unit length input vectors:

$$\mathbf{v_t} = \text{argmax}_{\mathbf{x}, ||\mathbf{x}||_2 = 1} ||B_t A_t \mathbf{x}||_2. \tag{4}$$

For this reason, Ostapenko et al. (2024) use $\mathbf{v}_t$ as the prototype for expert $t$, as inputs similar to $\mathbf{v}_t$ tend to produce activations with larger magnitudes. Let $P_\ell \in \mathbb{R}^{T \times d_{\text{in}}}$ be the routing matrix for layer $\ell$ with row $P_\ell[t] = \mathbf{v}_t$ representing the prototype for expert $t$. Ostapenko et al. (2024) then route an input vector $\mathbf{x}$ of layer $\ell$ to the top-$k$ experts using:

$$\text{experts} = \text{arg top-k}(|P_\ell \mathbf{x}|). \tag{5}$$

Ostapenko et al. (2024) use rank-4 LoRAs for their experiments, but we hypothesize that Arrow routing becomes less effective as the rank of experts increases. As the rank increases, the top eigenvector will capture a smaller percentage of the overall variation induced by that expert. We test this hypothesis and create an improved routing mechanism that leverages the entire spectrum of the LoRA covariance matrix without storing additional prototypes.

## 3  Spectral Routing

Spectral Routing (SPECTR) is our approach for dynamic token- and layer-wise composition of LoRAs, enabling improved multi-task adaptation of a base model without explicitly learning to route from data. SPECTR includes separate initialization and inference procedures.

---

[2]LoRA applies a fixed scalar to the matrix product, which we absorb into $B$ for cleaner notation.

### 3.1 Spectral Alignment

We use an identical initialization procedure for each layer of the base model. Let $B_t$ and $A_t$ be the current layer's LoRA parameters for expert $t$. We use the SVD from Equation 2 to compute $U_t$, $S_t$, and $V_t$. This can be done efficiently using low-rank or probabilistic algorithms (Halko et al., 2011; Nakatsukasa, 2019). Following from Equation 3, we reformulate the adapter parameters as:

$$B_t^* = U_t \text{ and } A_t^* = S_t V_t^T, \tag{6}$$

and discard the original parameters. Observe that $A_t^*$ contains scaled eigenvectors of the covariance matrix, the first of which is the prototype used in Arrow routing (Ostapenko et al., 2024). Instead of storing separate prototypes, we capture all covariance structure directly in the new adapter representation. We repeat this process for each layer and adapter.

### 3.2 Routing Procedure

Given a set of aligned adapters $\{(B_1^*, A_1^*), (B_2^*, A_2^*), \ldots, (B_T^*, A_T^*)\}$ and a token vector input to the current layer $\mathbf{x}$, we compute a low-rank representation $\mathbf{h}_t$ for adapter $t$ using:

$$\mathbf{h}_t = A_t^* \mathbf{x}. \tag{7}$$

Observe that if $A_t^*$ was rank-1, then $\mathbf{h}_t \equiv P_\ell[t]\mathbf{x}$ from Equation 5. SPECTR takes advantage of the full rank of $A_t^*$ and computes $\mathbf{h}_t$ directly from the aligned adapter parameters without requiring the separate prototype. We then compute a routing score for adapter $t$ using:

$$s_t = ||\mathbf{h}_t||_2, \tag{8}$$

which measures the length of $\mathbf{h}_t$ in the subspace induced by adapter $t$. Similar to Equation 4, we expect related vectors to maximize this score because the subspace represents directions of maximum variation for vectors used to train adapter $t$. We route to the top-$k$ experts with:

$$\text{experts} = \arg \text{top-k}_{t \in \{1, \ldots, T\}}(s_t). \tag{9}$$

### 3.3 Merging Procedure

SPECTR is flexible enough to allow for more advanced merging methods such as DARE (Yu et al., 2024), TIES (Yadav et al., 2023), or KnOTS (Stoica et al., 2025), but we use the same linear merging procedures as our baselines for a fair comparison. We discuss alternative merging options and considerations in Appendix A. Given the set of selected experts indexed $1, \ldots, k$, we uniformly average their low-rank representations from Equation 7:

$$\mathbf{h}^* = \frac{1}{k} \sum_{i=1}^{k} \mathbf{h}_i. \tag{10}$$

Similarly, we uniformly merge the selected experts remaining LoRA parameters:

$$\hat{B} = \frac{1}{k} \sum_{i=1}^{k} B_i^*. \tag{11}$$

Like Ostapenko et al. (2024), a softmax of the routing scores could be used as an alternative to uniform weights. Finally, we compute the output of the current layer for token $\mathbf{x}$ with:

$$\mathbf{x}^* = W\mathbf{x} + \hat{B}\mathbf{h}^*. \tag{12}$$

SPECTR performs these routing and merging procedures on a per-token and per-layer basis, enabling model expressivity on the scale of traditional MoE models.

## 4 Experiments

We measure the effectiveness of SPECTR as a routing strategy, testing our hypothesis that it improves routing accuracy over Arrow's rank-1 prototypes. We measure multi-task performance across different methods and explore the confounding impact of task similarity.

### 4.1 Models and Adapters

We replicate our experiments across four popular instruction-tuned LMs chosen for their diversity in parameter count: Gemma-2B[3] (Team et al., 2024), Phi-3.5B[4] (Abdin et al., 2024), and Llama-3B[5] and -8B[6] (Grattafiori et al., 2024). LoRA adapters (Hu et al., 2022) are trained for each model using the peft library (Mangrulkar et al., 2022). We apply rank-8 adapters to all linear layers of each model. A main motivation for SPECTR is to allow the use of externally sourced adapters, so we use default hyperparameters instead of optimizing them for each routing method. We prompt each model using the corresponding template from the huggingface library (Wolf et al., 2020) and include our training details in Appendix B. Keeping with Ostapenko et al. (2024), we use $k = 4$ for Arrow and SPECTR top-$k$ merging.

### 4.2 Datasets

We measure performance on several public datasets covering a diverse set of tasks: agnews,[7] cola (Warstadt et al., 2019), dbpedia (Auer et al., 2007), hellaswag (Zellers et al., 2019), mnli (Williams et al., 2018), mrpc (Dolan & Brockett, 2005), qnli (Rajpurkar et al., 2016), qqp,[8] and sst2 (Socher et al., 2013). We include each model's out-of-sample accuracy before and after fine-tuning on the datasets in Appendix C.

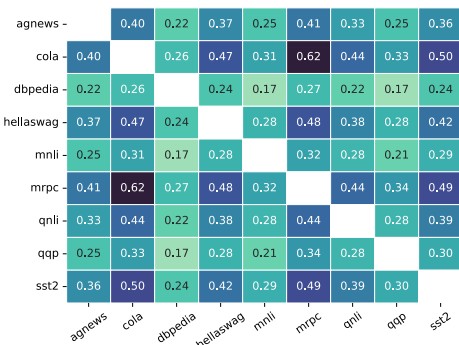

Figure 2: Pairwise cosine similarities between task adapters averaged across four models.

We measure the cosine similarity between the LoRA weights of different tasks as a proxy for task similarity. Figure 2 displays the pairwise similarity scores averaged over the four LMs. We include the cosine scores for individual models in Appendix D.

We compute an overall similarity score for each task by averaging its pairwise cosine scores (Figure 3). The score for each task is consistent across the four LMs, with a slight drop for Llama-8B likely due to its increased dimensionality. The dbpedia, mnli, and qqp adapters are the least similar to other tasks, while sst2, cola, and mrpc are the most similar.

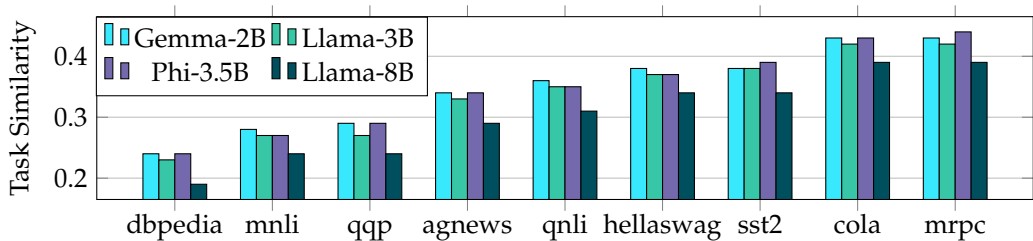

Figure 3: The average cosine similarity between a specific task and all other tasks.

### 4.3 Routing Performance

We first compare strategies by framing routing as a classification problem. Arrow and SPECTR both compute per-token routing scores at each layer. We label each token from an

---

[3] https://huggingface.co/google/gemma-2-2b-it
[4] https://huggingface.co/microsoft/Phi-3.5-mini-instruct
[5] https://huggingface.co/meta-llama/Llama-3.2-3B-Instruct
[6] https://huggingface.co/meta-llama/Meta-Llama-3-8B-Instruct
[7] http://groups.di.unipi.it/~gulli/AG_corpus_of_news_articles.html
[8] https://quoradata.quora.com/First-Quora-Dataset-Release-Question-Pairs

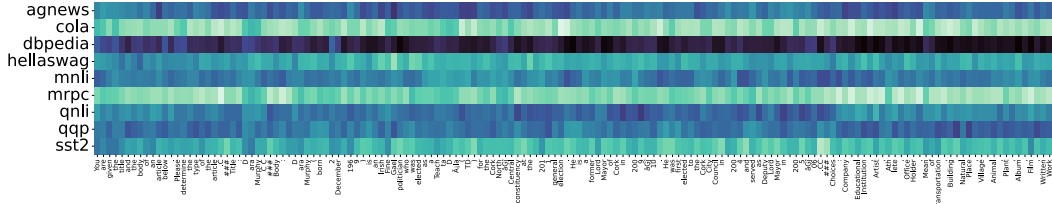

Figure 4: Per-token routing distribution using SPECTR for a sequence of tokens from dbpedia averaged across model layers. Darker areas indicate a higher average routing score.

out-of-sample sequence as belonging to the adapter trained on the corresponding dataset. We visualize the classification of an example sequence in Figure 4, where routing scores have been averaged across all model layers.[9] We note that a perfect routing accuracy at the token level would be unexpected, as only a subset of tokens are consistent across examples from the same dataset. We therefore focus on the relative difference in accuracies between the two approaches. We mark each token correct if the ground truth adapter is selected among the top-4 experts and record the routing accuracy in Table 1. SPECTR results in an average top-4 accuracy gain of approximately 4 percentage points, with individual task gains of up to 24 points. We include top-1 accuracies in Appendix E.

We plot routing accuracies against each task similarity score in Figure 5 and see that SPECTR's largest accuracy gains occur for the more unique tasks such as dbpedia. SPECTR seems to systematically choose alternative adapters for the most similar tasks such as mrpc and cola, resulting in accuracies below random chance. We will explore if these inaccuracies are consequential on downstream performance in Section 4.5, or if SPECTR's choices still perform well due to selected adapters having been trained for similar tasks. Arrow's accuracy appears independent of task similarity, albeit at a significantly lower average accuracy.

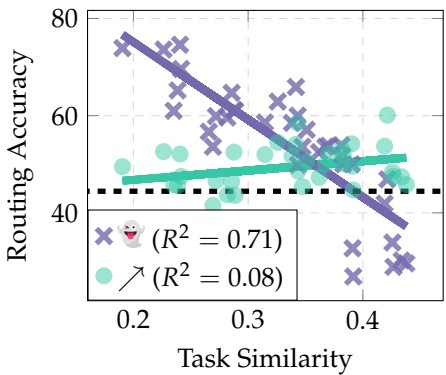

Figure 5: SpectR (🤖) is more accurate at routing than Arrow (↗) except for datasets with higher task similarity. The black dashed line indicates random routing accuracy.

| | Gemma-2B | | Llama-3B | | Phi-3.5B | | Llama-8B | |
|---|---|---|---|---|---|---|---|---|
| | ↗ | 🤖 | ↗ | 🤖 | ↗ | 🤖 | ↗ | 🤖 |
| agnews | 51.7 | 58.5 | 54.1 | 65.9 | 54.5 | 62.9 | 52.5 | 61.2 |
| cola | 48.4 | 33.9 | 47.4 | 30.2 | 53.7 | 41.9 | 49.6 | 32.7 |
| dbpedia | 47.5 | 69.6 | 52.1 | 74.6 | 52.6 | 73.6 | 49.5 | 73.9 |
| hswag | 50.1 | 51.8 | 52.2 | 53.8 | 50.0 | 53.6 | 48.6 | 49.9 |
| mnli | 43.7 | 59.6 | 46.6 | 59.9 | 38.0 | 56.7 | 45.8 | 61.0 |
| mrpc | 47.5 | 28.9 | 45.8 | 29.6 | 60.1 | 46.9 | 44.7 | 26.9 |
| qnli | 47.3 | 52.5 | 50.6 | 57.1 | 45.4 | 53.6 | 52.0 | 58.6 |
| qqp | 46.5 | 64.7 | 43.6 | 61.0 | 41.5 | 53.6 | 45.6 | 65.2 |
| sst2 | 51.8 | 50.0 | 54.2 | 50.0 | 53.1 | 53.9 | 58.5 | 60.0 |
| **AVG** | 48.3 | **52.2** | 49.6 | **53.6** | 49.9 | **55.2** | 49.6 | **54.4** |

Table 1: Top-4 routing accuracies. SPECTR(🤖) outperforms Arrow (↗) on all four models.

---

[9]We acknowledge the token labels are small, but keep them for curious readers.

### 4.4 Rank vs. Routing Effectiveness

We recall our hypothesis that Arrow prototypes become less meaningful as the adapter rank increases, reducing the sensitivity of routing scores to anything other than the single direction of maximum variation for each task. We quantify this effect by measuring the ratio of the routing score for the ground truth adapter against the highest routing score among all adapters. A ratio of 1 indicates perfect token routing. We plot these ratios for both methods as a function of adapter rank in Figure 6. The SPECTR routing scores from Equation 8 take advantage of the extra dimensions in the intermediate representation as the rank increases, leading to better discrimination between adapters. In contrast, Arrow uses the rank-1 prototype, which we confirm results in decreased ratio scores at higher ranks.

### 4.5 Multi-Task Performance

Now that we have established SPECTR improves routing, we confirm the strategy results in good multi-task performance in practice. We compare the base instruction-tuned model with Arrow and SPECTR, and we include $\mu$-routing as an additional strong baseline. Following similar work, we measure the normalized accuracy to control for the difficulty of the task (Yadav et al., 2023; Ilharco et al., 2023; Stoica et al., 2025). Normalized accuracy divides the achieved performance by the accuracy of the oracle model fine-tuned for the specific task.

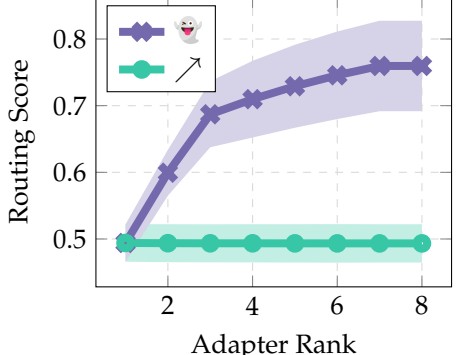

Figure 6: SPECTR(🤡) becomes increasingly more effective than Arrow (↗) at selecting the correct adapter as adapter rank increases.

We report the normalized accuracies in Table 2. All three routing methods outperform the baseline models and SPECTR results in the highest average accuracy in all cases. The magnitude of individual performance differences varied across models and tasks, with SPECTR's largest gain over $\mu$ and Arrow being 15.1 and 10.7 percentage points respectively.

SPECTR performed well on tasks even when routing accuracy was low, suggesting that low routing accuracies were partially caused by different datasets having similar tasks. High task similarity resulted in good performance even though the ground truth adapter was used less often. Interestingly, all routing methods and LMs perform poorly on hellaswag, even though the individual adapters fine-tuned for that task do well (Appendix C). This could indicate task interference (Ortiz-Jimenez et al., 2023; Tang et al., 2024), possibly caused by the simple linear merge used by all three methods in our experiments (Appendix A).

## 5 Trade-offs

While better routing and multi-task performance are desirable, there is no free lunch when choosing between methods (Wolpert & Macready, 1997). We discussed the impact of adapter rank in Section 4.4, and here we highlight three additional dimensions to consider when choosing between training-free strategies: storage cost, VRAM usage, and interference.

**Storage** costs for $\mu$-routing and SPECTR are identical if the experts are kept isolated for circumstances such as access-control (Fleshman et al., 2024). Otherwise, $\mu$-routing can have zero overhead by merging all experts into the base model ahead of time. SPECTR represents the experts in a different form after spectral alignment (Section 3.1), but the resulting matrices are the same size and produce equivalent products. Arrow routing requires the same library but includes the additional overhead of storing prototype vectors for each expert at every layer in the model (Ostapenko et al., 2024). The additional overhead is equivalent to storing an extra adapter for every $2r$ experts when using LoRAs of rank $r$.

|  |  | agnews | cola | dbpedia | hswag | mnli | mrpc | qnli | qqp | sst2 | **AVG** |
|---|---|---|---|---|---|---|---|---|---|---|---|
| **Gem-2B** | Base | 86.2 | 92.7 | 88.4 | 38.3 | 62.2 | 90.4 | 73.2 | 84.9 | 95.0 | 79.0 |
|  | $\mu$ | 87.6 | 93.2 | 95.4 | 33.1 | 69.9 | 89.1 | 60.8 | 90.1 | 96.9 | 79.6 |
|  | ↗ | 87.9 | 92.3 | 95.2 | 29.8 | 53.2 | 91.9 | 53.1 | 91.4 | 96.6 | 76.8 |
|  | 👻 | 88.2 | 91.3 | 96.8 | 29.7 | 69.8 | 90.8 | 63.8 | 92.0 | 96.7 | **79.9** |
| **Llam-3B** | Base | 79.7 | 65.3 | 86.1 | 39.0 | 38.0 | 63.3 | 72.9 | 53.5 | 88.3 | 65.1 |
|  | $\mu$ | 84.3 | 90.7 | 94.1 | 53.0 | 55.6 | 86.4 | 83.2 | 82.9 | 95.7 | 80.7 |
|  | ↗ | 85.3 | 88.2 | 95.1 | 53.9 | 49.7 | 84.6 | 83.8 | 90.4 | 94.4 | 80.6 |
|  | 👻 | 84.8 | 89.4 | 96.1 | 52.4 | 55.3 | 87.4 | 83.3 | 91.9 | 95.7 | **81.8** |
| **Phi-3.5B** | Base | 78.9 | 95.0 | 94.4 | 65.6 | 80.6 | 94.1 | 60.8 | 86.5 | 94.0 | 83.3 |
|  | $\mu$ | 84.8 | 96.2 | 97.3 | 69.3 | 86.3 | 94.0 | 59.2 | 91.9 | 95.7 | 86.1 |
|  | ↗ | 85.0 | 96.2 | 97.7 | 73.9 | 88.2 | 92.5 | 56.9 | 92.1 | 95.9 | 86.5 |
|  | 👻 | 86.0 | 96.2 | 98.2 | 74.3 | 88.3 | 94.0 | 55.9 | 92.4 | 95.8 | **86.8** |
| **Llam-8B** | Base | 88.1 | 84.5 | 94.0 | 52.4 | 57.3 | 91.9 | 81.3 | 75.9 | 93.4 | 79.9 |
|  | $\mu$ | 93.3 | 87.4 | 99.1 | 41.5 | 73.0 | 91.8 | 90.2 | 94.5 | 97.7 | 85.3 |
|  | ↗ | 90.2 | 93.5 | 97.0 | 56.4 | 71.0 | 93.8 | 89.0 | 91.4 | 97.2 | 86.6 |
|  | 👻 | 88.4 | 93.2 | 98.0 | 56.6 | 72.2 | 93.8 | 90.2 | 92.7 | 97.3 | **86.9** |

Table 2: Multi-Task performance across models and methods.

**VRAM** requirements are lowest for $\mu$-routing because inputs are processed by a single merged adapter. Arrow and SPECTR require that all adapters be loaded in GPU memory as selection and merging occur on the fly for each token at each layer. Arrow takes advantage of its rank-1 prototypes to reduce computation in the routing step, while SPECTR requires all adapters to compute low-rank representations before selecting the top-k experts. For a layer with hidden dimension $h$, routing with SPECTR requires the FLOPS equivalent to a forward pass of the layer for every $h/r$ experts. Arrow allows for $h$ experts at the same cost.

**Interference** between adapters is most likely when using the dense selection of $\mu$-routing (Caccia et al., 2023; Ostapenko et al., 2024). Both Arrow and SPECTR help mitigate interference by using sparse routing and are compatible with additional mitigations such as DARE (Yu et al., 2024), TIES (Yadav et al., 2023), or KnOTS (Stoica et al., 2025) if necessary.

These considerations can help practitioners choose a strategy under various constraints. Our experiments demonstrate superior performance with SPECTR, but Arrow might be preferable for its increased GPU efficiency as long as the adapters are of low enough rank. $\mu$-routing remains a good option for smaller adapter libraries where routing is unnecessary.

## 6 Conclusion

In this work, we studied training-free model MoErging approaches for integrating externally trained LoRA experts into expressive models capable of performing well in a multi-task environment. We identified limitations with existing strategies, such as interference and sensitivity to adapter rank. These challenges led to our development of SPECTR, a new approach for per-token, per-layer routing, requiring zero additional data or training to employ. We conducted experimentation across a diverse set of tasks and popular LMs, demonstrating the ability of SPECTR to leverage the extra dimensions of higher rank adapters for increased routing accuracy over alternatives. SPECTR improved routing accuracy by 4 percentage points on average, leading to individual increases in task performance of up to 15%. We also explored the impact of task similarity on routing accuracy, and we found the results for each dataset were consistent across all four LMs under evaluation. SPECTR achieved the highest accuracies on the more unique tasks. We found that the alternate adapters selected by SPECTR for highly similar tasks still performed well regardless of imprecise routing. Finally, we discussed trade-offs to consider when choosing a routing strategy, including storage costs, VRAM requirements, and potential adapter interference. Overall, we hope the effectiveness of SPECTR will allow for broader use of the abundance of expert models available for composition and improve the multi-task capabilities and performance of LMs.

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

## A  Alternate Merging Procedures

As mentioned in Section 3.3, we structure Equation 10 and Equation 11 to make comparisons with baselines more fair. However, merging the experts in two separate steps is known to cause issues when the adapters were initialized with different seeds (Chronopoulou et al., 2023; Fleshman et al., 2024), as the merging procedure is sensitive to the ordering of rows and columns in each matrix. The issue is alleviated by merging the final result of each expert instead of doing separate merges:

$$\hat{\mathbf{x}} = \frac{1}{k} \sum_{i=1}^{k} B_i^* \mathbf{h}_i,$$ (13)

and using $\hat{x}$ in place of $\hat{B}\mathbf{h}^*$ in Equation 12. This removes the need for experts to have the same permutation of rows and columns and allows for seamless merging of adapters with different ranks, as the resulting products will have the same dimensions. This simple average is also easily generalized to more sophisticated merging algorithms.

## B  LoRA Training

All adapters were trained on a single Nvidia A100 GPU with 80GB of memory. We used rank-8 LoRAs with a LoRA $\alpha = 16$, LoRA dropout of 0.05, and applied adapters to all linear layers. We used supervised fine-tuning and trained with a batch size of 8 using a learning rate of 1e-4 with a constant learning rate scheduler.

## C  Adapter Performance

We computed the raw out-of-sample accuracy achieved by each model before and after fine-tuning the model on the in-sample portion of each dataset (Table 3).

|         | Gemma-2B Base | FT   | Llama-3B Base | FT   | Phi-3.5B Base | FT   | Llama-8B Base | FT   |
|---------|------|------|------|------|------|------|------|------|
| agnews  | 80.6 | 93.5 | 75.3 | 94.5 | 74.9 | 94.9 | 83.0 | 94.2 |
| cola    | 79.0 | 85.2 | 55.5 | 85.0 | 81.6 | 85.9 | 73.7 | 87.2 |
| dbpedia | 87.5 | 99.0 | 85.2 | 99.0 | 93.6 | 99.2 | 93.2 | 99.1 |
| hswag   | 34.4 | 89.9 | 34.2 | 87.6 | 60.1 | 91.6 | 48.3 | 92.2 |
| mnli    | 54.8 | 88.1 | 33.8 | 88.9 | 72.8 | 90.3 | 51.5 | 89.9 |
| mrpc    | 75.7 | 83.7 | 54.1 | 85.5 | 77.8 | 82.7 | 75.0 | 81.6 |
| qnli    | 67.5 | 92.2 | 67.5 | 92.6 | 56.5 | 93.0 | 75.6 | 93.0 |
| qqp     | 75.7 | 89.2 | 47.8 | 89.4 | 77.5 | 89.6 | 68.2 | 89.9 |
| sst2    | 90.2 | 94.9 | 84.1 | 95.2 | 89.9 | 95.6 | 88.7 | 95.0 |

Table 3: Task Performances before and after fine-tuning a task-specific adapter.

## D  Task Similarities

We display the cosine similarities between adapters for all models in Figure 7.

## E  Top-1 Routing

We show the top-1 routing accuracies on out-of-sample data in Table 4.

|         | Gemma-2B ↗ | 👻   | Llama-3B ↗ | 👻   | Phi-3.5B ↗ | 👻   | Llama-8B ↗ | 👻   |
|---------|------|------|------|------|------|------|------|------|
| agnews  | 18.4 | 21.9 | 20.5 | 27.0 | 20.5 | 25.9 | 19.2 | 23.7 |
| cola    | 14.0 | 7.5  | 13.2 | 7.0  | 14.7 | 11.1 | 14.2 | 7.1  |
| dbpedia | 16.1 | 36.0 | 19.7 | 39.6 | 20.0 | 41.6 | 16.6 | 38.5 |
| hswag   | 17.2 | 17.8 | 18.7 | 19.2 | 19.3 | 21.5 | 16.3 | 15.7 |
| mnli    | 11.0 | 20.3 | 12.8 | 21.6 | 10.5 | 21.3 | 12.5 | 22.1 |
| mrpc    | 11.6 | 4.9  | 12.5 | 6.0  | 18.0 | 11.4 | 11.4 | 5.6  |
| qnli    | 13.4 | 14.4 | 14.0 | 15.6 | 14.4 | 17.2 | 15.6 | 17.4 |
| qqp     | 15.9 | 26.3 | 11.3 | 21.8 | 12.9 | 20.4 | 12.8 | 26.1 |
| sst2    | 16.3 | 15.4 | 18.1 | 14.7 | 17.8 | 19.1 | 20.3 | 19.4 |
| **AVG** | 14.9 | **18.3** | 15.6 | **19.2** | 16.5 | **21.1** | 15.4 | **19.5** |

Table 4: Top-1 routing accuracies.

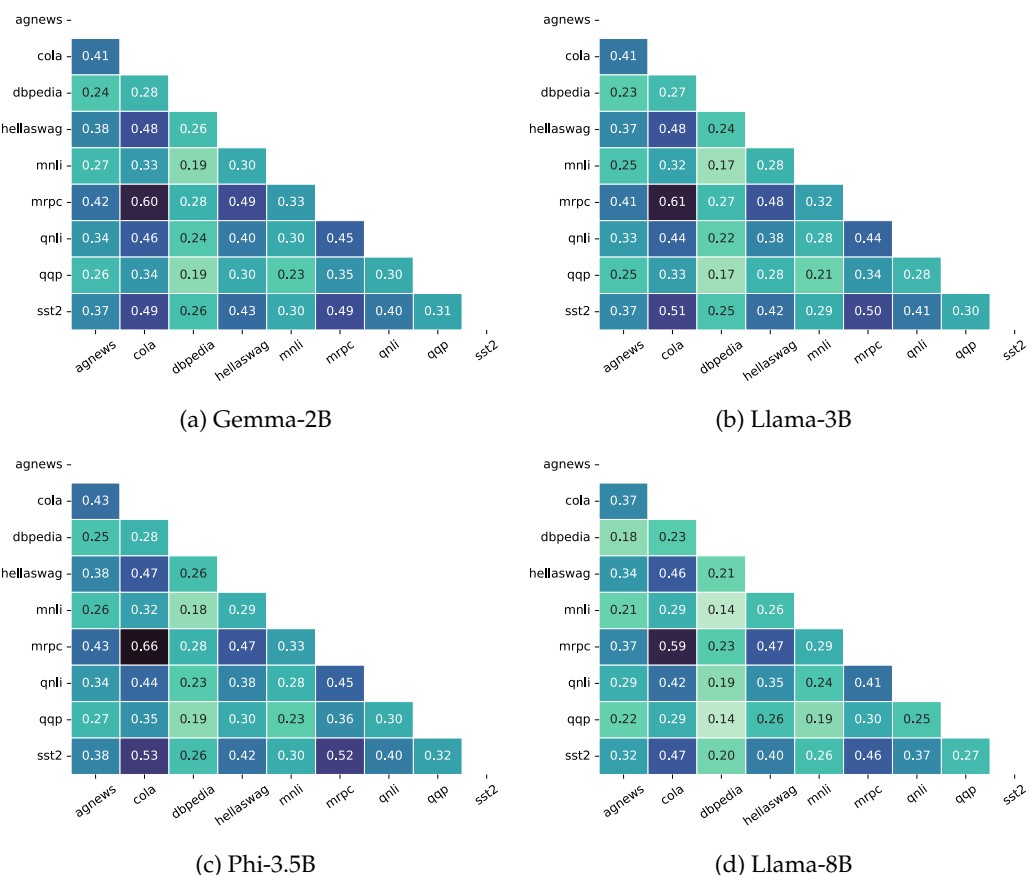

Figure 7: Cosine Similarities for all models.

