# OpenReview forum: "SpectR: Dynamically Composing LM Experts with Spectral Routing"
_colmweb.org/COLM/2025/Conference — COLM 2025_

### Official Review · Reviewer_Efhf · 2025-05-12

**Rating:** 6
**Confidence:** 4
**Ethics Flag:** 1

**Summary:**

This paper introduces SpectR, a training-free framework for enhancing multi-task performance in language models (LMs) through dynamic routing and merging mechanisms for task-specific LoRA modules during inference. Unlike existing approaches like Arrow routing, which relies on rank-1 prototypes derived from singular value decomposition (SVD) of LoRA parameters, SpectR leverages the full spectrum of the LoRA covariance matrix to compute routing scores. This enables token- and layer-wise expert selection while avoiding interference between adapters. Results show that SPECTR improves routing accuracy by 4 percentage points on average, leading to individual increases in task performance of up to 15%.

**Questions To Authors:**

It might be better to quantify practical trade-offs, including metrics like inference latency, VRAM usage, and scaling curves (experts vs. performance) to guide deployment strategies.

**Reasons To Accept:**

1. The paper introduces SpectR, a training-free method that employs the full spectrum LoRA covariance matrices via SVD. It overcomes the limitations of rank-1 prototype-based methods (e.g., Arrow) and improves routing accuracy/task performance through enhanced input variation modeling.
2. It explores the impact of adapter rank and task similarity on routing efficacy, advancing understanding of expert composition dynamics in LMs.
3. The method is compatible with interference-mitigation techniques (e.g., DARE, TIES), enhancing robustness for large-scale expert libraries.

**Reasons To Reject:**

1.	The computational cost of per-adapter/layer SVD decomposition remains unmeasured, with critical metrics (e.g., inference latency, memory footprint, scaling behavior across ranks/expert counts) absent, limiting practical feasibility analysis.
2.	SpectR’s performance with large-scale expert libraries (e.g., hundreds of adapters) and cross-domain tasks is untested, leaving scalability under unexplored scenarios unaddressed.
3.	Experiments are limited to text classification and inference tasks (e.g., MNLI, SST-2). Performance on diverse task types (e.g., generation, reasoning) remains unverified, raising questions about SpectR’s applicability beyond the tested domains.

---

> ### Author Response · Authors · 2025-05-28
>
> Thank you for taking the time to review and for considering our discussion of your concerns.
>
> 1. The SVD itself is of little concern as it computed once offline using efficient approaches (line 160). The SVD aligns the LoRA adapter, but it does not change its memory footprint or individual compute requirements.  The same SVD is required for our Arrow baseline.
>
> 2. This paper focuses on demonstrating that SpectR solves shortcomings in the rank-1 prototypes of Arrow routing using a similar experimental setting. Our future work involves large LoRA libraries, but this was not the focus of the Arrow approach we improve upon in this work.
>
> 3. We conduct experiments across 36 combinations of models and datasets, but agree that more models and datasets would be interesting to explore, especially across different modalities.
>
> 4. We devote Section 5 to discussing trade-offs, including quantifying the storage and VRAM costs as a function of the number of experts.

---

> > ### Comment · Reviewer_Efhf · 2025-06-10
> >
> > I appreciate the authors' clarification and will maintain my original score. Thanks

---

### Official Review · Reviewer_iGcM · 2025-05-18

**Rating:** 6
**Confidence:** 3
**Ethics Flag:** 2

**Summary:**

This paper addresses efficient deployment of LLMS through a training-free routing method—Spectral Routing (SPECTR)—for merging LoRA adapters without reliance on additional data or custom training procedures. The authors motivate their method by pointing out a key challenge of the existing method called Arrow Routing. By leveraging spectral properties of each adapter’s low-rank update, SPECTR constructs prototype vectors that more accurately capture expert relevance, resulting in consistent routing accuracy improvements over Arrow.

**Questions To Authors:**

1. Some discussion on whether this method can be extended to problems with lesser constraints. For e.g. one of the constraints in the problem statement is that we know that a query is related to task t, what there is not a direct query to task 1:1 matching but instead,  we have multiple tasks for each query and we need to increase top-k, will the method still outperform Arrow?

2. Not a question, but a suggestion - a lot of the paper space is used up just to describe related work, motivation and introduction. You could reduce that space and instead focus more on expanding the experiments on the need for higher rank in LoRA adapters.

**Reasons To Accept:**

The paper is well written with a good review of the existing methods and the motivation building up to the problem statement is easy to follow. The flow of the paper is also to the point. The problem statement is clearly motivated and the proposed method has been shown to outperform Arrow Routing which has a restriction on the top-eigenvector being chosen the element for the routing decision variable.

1. The proposed method instead uses the right singular vector as the element for the routing decision. Since it is full rank, it circumvents the issue in Arrow.
2. The results clearly show that on average, Spectr outperforms Arrow except for datasets with higher task similarity. This is a good finding.
3. The method is fairly simple to follow and easy to implement.

**Reasons To Reject:**

My main concern with the paper is two fold:

1. Can the authors discuss when would be need more ranks in the Arrow method and why is rank-4 not enough. That is a key question that needs to be empirically studied. I can understand that higher rank is a disadvantage in Arrow, but do we need more?

2. Following the above point, the authors need to do more robust experiments on showing how their method outperforms the Arrow routing on tasks. I see Section 4.4 discusses one example, but this study needs to be segregated for tasks. When is high rank necessary for Arrow, can we keep low rank and instead use larger topK?

---

> ### Author Response · Authors · 2025-05-28
>
> Thank you for taking the time to review and for considering our response to your questions and concerns.
>
> Addressing concerns:
>
> 1. The rank required for a LoRA expert is independent of Arrow or SpectR. LoRA trades off memory for performance. Task performance generally increases with rank, but the parameter efficiency decreases. Given a set of LoRA adapters, we show that SpectR is more effective than Arrow even at smaller ranks, and the relative gain from SpectR increases even more as rank increases (Figure 6). It is common for practitioners to train LoRAs with ranks even higher than what is explored in this paper.
>
> 2. We compare Arrow to SpectR across 4 models and 9 datasets. We separate the routing performance by task in Table 1. In general, the performance of our models would improve with higher rank, as the models have more capacity to learn. Our experiments show that at a given rank SpectR will outperform (Table 2) and SpectR’s effectiveness increases with rank (Figure 6). Using lower ranks and higher top-k would decrease performance on two fronts: the lower rank would reduce model capacity to learn new tasks and the higher k would increase interference between adapters (line 92+).
>
> Addressing questions:
>
> 1. We evaluate the model on tasks associated with the trained adapters, but we do not give the model information about which task a query belongs. We discuss several related works demonstrating that merging small numbers of LoRAs can lead to good performance in and out of domain (Section 2.2) and focus our work on training-free approaches to routing.
>
> 2. Different aspects of LoRA (including rank) have been heavily discussed in other works. We hope our previous comments have mitigated this concern for you.

---

### Official Review · Reviewer_N8tG · 2025-05-19

**Rating:** 6
**Confidence:** 4
**Ethics Flag:** 1

**Summary:**

This paper presents SpectR (Spectral Routing), a training-free approach during inference for dynamically composing (MoErging) multiple task-specific LoRA fine-tuned variants (experts) of the same base LM. The key innovation lies in using SVD of LoRA matrices to create low-rank spectral representations that, as the authors claim, can equivalently measure experts' input compatibility without requiring additional training data, or introducing prototypes as in Arrow routing. Based on $\mathcal{L}^2$ norm of these representations as routing scores, SpectR then averages outputs of the top-k most compatible experts at both per-token and per-layer granularities.

The authors demonstrate improvements on SpectR's average routing accuracy and performance when compared with Arrow routing, and how the effectiveness of SpectR scales better with adapter rank compared to Arrow routing. The authors provide additional discussions with valuable insights into the relationship between different methods' performance and efficiency considerations.

**Questions To Authors:**

+ In Section 3.3 line 183 the authors suggest a **softmax of routing scores** could be used to make a weighted average of  selected Top-K experts, which is logically intuitive and has indeed been widely adopted in the routing process of modern MoE LLM architectures. This should have become an important design choice for SpectR, as distribute weights to experts with higher scores would easily outperform uniform distribution. Yet it is **never experimented / discussed** in the following sections. Could you elaborate on why the simple uniform averaging works well for merging selected experts? Have you experimented with weighted averaging based on routing scores?

+ The paper provides **no ablation studies**. It would be more solid if the authors could assess the specific accuracy/performance gain to see the contribution of each components of the proposed methodology, for example, comparing the introduction of SVD over other simpler alternatives like a direct $\mathcal{L}^2$ norm on the intermediate hidden state between LoRA matrices [1].

+ In Section 4.2, cosine similarities between LoRA matrices for different tasks are used as indicator of task similarities. However the paper fail to specify whether during their fine-tuning process, for a same base LLM, the **LoRA adapters for different tasks are always initialized with identical weights**. It is stated in Appendix A that a initialization with different seeds would bring trouble to merging matrices. Nevertheless, when initialization differs across tasks, the observed cosine similarities may reflect initialization artifacts rather than genuine task relationships, which may fundamentally undermines the task similarity analysis and its conclusions about SpectR's performance patterns in Sections 4.3 & 4.5.

+ In Table 1&2, the results focused on top-4 routing, and in appendices only top-1 is provided. These **choices of top-k** appears arbitrary without justification or sensitivity analysis across different k values. Is there a principled way to determine the optimal k for a given set of experts? What happens when the number of available experts increases, does the routing accuracy degrade, and are there scalability concerns? Adding related experiments and demonstrate results in plots would bring valuable practical insights on this critical scaling factor for deploying SpectR.



### References
[1] Lv, Ang, Ruobing Xie, Yining Qian, Songhao Wu, Xingwu Sun, Zhanhui Kang, Di Wang, and Rui Yan. "Autonomy-of-Experts Models." arXiv:2501.13074 (2025).

**Reasons To Accept:**

+ The technical approach in this paper is novel. Their use of full SVD spectrum instead of just the top eigenvector (as in Arrow routing) is a meaningful technical contribution that, as the authors demonstrate, can better leverages the rank structure of LoRA adapters.
+ The experimental validation is substantial, with extensive fine-tuning and MoErging experiments across 4 different LLMs and 9 diverse tasks, demonstrating improvements of SpectR in routing accuracy and performance.
+ The training-free nature of SpectR makes it highly applicable for real-world scenarios, and the authors also provide valuable discussions with insights that could facilitate practical deployments.
+ The paper is well-written with clear structure and nice visual aids that effectively communicate their approach.

**Reasons To Reject:**

### 1. Limited Scope and Generalizability
There is a **significant mismatch between the claims made and the currently presented contents**, which may mislead readers about the method's broader applicability and impact.
The paper's title and abstract claim to address "dynamically composing LM experts" and refer broadly to "specialized expert models, fine-tuned from pretrained models," suggesting a general framework for expert composition. However, the methodology and experiments are entirely focused on LoRA adapters without exploration of generalizability to other forms of LM experts based on full model fine-tuning or other PEFT methods.  The authors should either reformulate the scope to directly address LoRA-based experts, or provide additional validation across different expert types to support the general claims.

### 2. Insufficient Theoretical Foundation
The paper would benefit from deeper theoretical analysis of why the spectrum approach should work better than Arrow routing or any other potential designs. Current explanation for why full spectrum outperforms Arrow's rank-1 prototypes remains largely intuitive rather than rigorously grounded.

### 3. Marginal and Inconsistent Improvement
As is demonstrated in Table 1&2, SpectR's improvement against Arrow routing and $\mu$-routing is rather marginal and task-dependent (as discussed, suffering from task similarities). Based on performance reported in Table 2, the improvement of SpectR against $\mu$-routing baseline is not statistically significant (with p=0.12 from paired t-tests, and win rate only 19/36=53%), and the win rate against Arrow routing is still only 24/36. These brings concerning limitations to the practical significance of methods proposed.

### 4. Missing Details
The paper lacks several critical information and experiment results that could undermine the core claims made in the paper. Please find the questions raised in the following "Questions To Authors" sections.

---

> ### Author Response · Authors · 2025-05-28
>
> Thank you for the detailed review. We attempt to mitigate your stated concerns and answer your questions below.
>
> 1. Scope and Generalizability
>
> We discuss existing work showing that alternative finetuning strategies can be converted into LoRAs using the difference in weights and state that we focus on only traditional LoRA models (Line 83). However, we can make it clear in the abstract that the experiments use only LoRA-based experts which make up a significant share of expert fine-tunes in model repositories such as huggingface.
>
> 2. Theoretical Foundation
>
> We ground our experimental findings in the spectral theory covered in Section 2.4, namely that the right singular vectors (those used by SpectR) are the “orthogonal directions of maximum variation induced by the LoRA expert in the space of input vectors”. Unlike Arrow, “we capture all covariance structure directly in the new adapter representation” (165) which intuitively leads to better performance as seen by the experimental results.
>
> 3. Improvement
>
> Our improvement over Arrow routing is statistically significant with a paired t-test on Table 2 producing p=0.036. While μ-routing is a strong baseline for a smaller number of tasks, existing work shows that as the number of tasks grow, interference becomes a significant problem (line 43). With only 9 tasks we still outperform μ-routing on average across all four models, although the gain is less significant.
>
> Answers to Questions
>
> 1. The top-k selection distributes higher weights to higher scores by giving zero weight to the lowest scoring experts. We did explore softmax vs uniform weighting of the selection and found the results to be very similar. The vector norms are discriminative but their variance is small, leading to softmax outputs that are essentially uniform already. We originally chose a uniform weighting to encourage a more stable distribution of experts across layers while also aware of the previous work showing uniform merges work well with smaller k. This ended up not being a major factor, as the routing is similar either way. We can include more discussion of this in an appendix, but the main body details the chosen approach which was used for all our experiments.
>
> 2. Our baselines effectively ablate sparse vs. dense (mu) routing and rank-1 (Arrow) vs full rank prototypes. Our problem statement is specific to training-free approaches which isn’t applicable to the referenced paper which trains models to produce larger activations for the correct expert.
>
> 3. All adapters are initialized with the same weights. We will add a clarifying sentence to our LoRA training details.
>
> 4. We choose top-4 following the original Arrow paper to make comparisons with the previous work easier. The value of k could be tuned with out-of-sample data but this would break our training-free goal. Previous work has shown that smaller k merges are better due to increased interference at larger k. We do not experiment with a larger number of tasks in this work, but have positive results in our follow-up work.

---

> > ### Comment · Reviewer_N8tG · 2025-05-28
> >
> > Thank you for the response addressing my concerns. The authors' reply cleared up some of my doubts. I believe the authors have designed solid methodology and gives experiments with interesting results. There are still several part I am not quite sure about:
> > + **Regarding softmax for weighted outputs.**
> >      + In the authors' reply they stated that "The vector norms are discriminative" (sufficient for meaningful top-k selection), and "Their variance is small, leading to softmax outputs that are essentially uniform". This makes sense to me, and I believe it's actually very intuitively explainable & fixable.
> >     + I do agree with the authors' claim that the norms as routing scores should be discriminative. But by the given nature of the task and experimental setups, there should be at least one clear expert differentiation among the selected top-k ones: when evaluating on each dataset, there should always be 1 expert that is trained exactly on that dataset, yet with current method it (if it is activated among the top-4) would be treated with same weights as the others. So a proper utilization of their norms really should further boost the performance beyond uniform distribution.
> >     + Consider that traditional MoE routers use routing scores from high-dimensional vector multiplication, which (let's assume that weights & hidden states were randomly initialized) would yield an expectation around 0, where softmax works well. However, SpectR is based on routing scores from $\mathcal{L}^2$ norm values on low rank vector, which do not take negative values and would concentrate around a positive expected value (~$\sqrt{rank}$ maybe), hence introduce a basic magnitude level that leads to the observed near-uniform uniform softmax results.
> >     + An intuitive implementation is subtracting a constant expected $\mathcal{L}^2$ norm for random rank-r matrices or an average value calculated from that batch before using softmax.
> > + **Regarding Ablation**:
> >     + I agree that $\mu$ routing and Arrow routing serve as meaningful baselines, but I still think a more granular ablation studies would help significantly strengthen the paper by examining each individual component's contribution to SpectR's performance.
> >     + For example, a comparison of SpectR against computing vector norm directly on the intermediate between original LoRA matrices without SVD (structurally similar to [1], the reason I've cited it) can isolate how much performance does the SVD add beyond simple matrix norms. This can also be done for different SVD components, like using norms on the vector after $U_t$ without $V_t$ for routing.
> >     + This kind of ablation, that experiments which components contribute how much of the performance, is very easy to implement (only on smallest LM is sufficient) and will facilitate a much deeper understanding of SpectR with more valuable insights in this matter. In the current version of this paper, it's unclear whether improvements come from the SVD transformation, the specific spectral representation, or simply using better norms than existing methods.
> >
> > Considering these factors, I decide to raise my score to 6 for now. I believe the authors could and should continue devoting some efforts to make this work much better.

---

> > > ### Author Response · Authors · 2025-05-29
> > >
> > > We really appreciate your feedback.
> > >
> > > Although there is a single adapter trained on the task related to an evaluated sequence, the routing occurs on a token-by-token basis. Our comments starting at line 227 discuss that we shouldn't expect perfect accuracy, as for any given token another adapter could be more suitable. Using top-4 increases the frequency of selecting the ground-truth adapter, and the uniform weighting ensures the adapter contributes the same amount regardless of its relative position. Experts at a particular layer in a traditional MoE are trained to receive vectors from any experts from the previous layers. Our LoRA-based experts have only seen vectors produced by the corresponding adapter. This is why we thought uniform weighting might be preferable for maintaining a similar mixture of the selected adapters across the sequence and layers.
> > >
> > > Softmax is translation invariant, so subtracting a constant wouldn't change the result. A temperature scaling factor could be introduced to get a peakier distribution, but tuning that hyperparameter would break our training-free goals.
> > >
> > > Thank you again.

---

### Decision · Program_Chairs · 2025-07-08

**Decision:**

Accept

**Comment:**

This paper proposes SpectR, a training-free method that dynamically composes LoRA-based expert models using spectral properties of adapter matrices during inference. The approach improves upon Arrow routing in both routing accuracy and task performance. Reviewers appreciated the clear motivation, technical contributions, and empirical validation across models and tasks. However, all reviewers raised concerns about limited generalizability beyond LoRA, which limits the method's practical value. While the authors provide some justifications, some concerns remain. Overall I lean toward accepting the paper, but only marginally.